# Feature Selection in High-Dimensional Models via EBIC with Energy Distance Correlation

**DOI:** 10.3390/e25010014

**Published:** 2022-12-21

**Authors:** Isaac Xoese Ocloo, Hanfeng Chen

**Affiliations:** 1Department of Statistics, University of Georgia, Athens, GA 30602, USA; 2Department of Mathematics and Statistics, Bowling Green State University, Bowling Green, OH 43403, USA

**Keywords:** energy distance, extended Bayesian information criteria, feature variable selection

## Abstract

In this paper, the LASSO method with extended Bayesian information criteria (EBIC) for feature selection in high-dimensional models is studied. We propose the use of the energy distance correlation in place of the ordinary correlation coefficient to measure the dependence of two variables. The energy distance correlation detects linear and non-linear association between two variables, unlike the ordinary correlation coefficient, which detects only linear association. EBIC is adopted as the stopping criterion. It is shown that the new method is more powerful than Luo and Chen’s method for feature selection. This is demonstrated by simulation studies and illustrated by a real-life example. It is also proved that the new algorithm is selection-consistent.

## 1. Introduction

Advancements in technology have led to the production of sophisticated machines which are able to measure many details about every observational or experimental unit in a system. This results in data with more features (*p* or predictors) than the number of observational or experimental units (sample size *n*), referred to as high-dimensional data. Most of these data come from genetic research, e-commerce, biomedical imaging, and functional magnetic resonance imaging, among many others.

Since there are many more features (*p*) than the sample size (*n*), they are analyzed using a sparse high-dimensional regression (SHR) model:(1)$$y_{i}=\beta_{0}+\sum_{j=1}^{p}x_{ij}\beta_{j}+\epsilon_{i},\;\mathrm{for}\;i=1,\ldots ,n.$$
It is assumed that there is only a relatively small number of the nonzero $\beta_{j}$’s. The main goal in their analysis is feature selection. As stated by [1], feature selection typically has two goals. The first is for model building using desirable prediction properties. The second is for identifying the features with nonzero coefficients. For convenience, such features are referred to as relevant features in this paper.

One approach to the SHR model is to estimate the $\beta_{j}$’s by a regularization method, which is done by simultaneously minimizing the penalized least squares below:$$\sum_{i=1}^{n}{\left(y_{i}-\beta_{0}-\sum_{j=1}^{p}\beta_{j}x_{ij}\right)}^{2}+\sum_{j=1}^{p}p_{\lambda }\left(\right|\beta_{j}\left|\right),$$
where λ is the regulating parameter and $p_{\lambda }$ is a penalty function. When $p_{\lambda }$ is based on the $L_{1}$ norm, thus ${∥\beta ∥}_{1}=\sum_{j=1}^{p}\left|\beta_{j}\right|$, it is referred to as the LASSO [2]. The $L_{1}$ norm penalty is able to shrink the coefficients of redundant predictors to zero. Thus, the LASSO usually results in sparse models that are easier to interpret. Other penalty functions such as the SCAD [3] and adaptive LASSO (ALasso) [4] have also been reported in the literature. SCAD smoothly clips the $L_{1}$ penalty (for small $|\beta_{j}|$), and assigns a constant penalty (for large $|\beta_{j}|$s). On the other hand, adaptive LASSO utilizes the minimax concave penalty [5], $p_{\lambda }\left(|\beta_{j}|\right)=\lambda w_{j}|\beta_{j}|$, where $w_{j}$ represents the weights. The regulating parameter, λ, is usually chosen using cross-validation (CV).

Another approach to analyzing the SHR model, large-*p*-small-*n* problem, is sequential variable selection, which is designed to reduce the dimension of the data such that d<p. There are two forms of sequential variable selection. The first uses the sure screening property, which selects from the many features a subset which contains the relevant features (predictors). This is usually followed by a regularization method such as SCAD or ALasso to identify and estimate the relevant predictors from the reduced feature space. The other form is to sequentially select the relevant features through a repetitive process which terminates when a stopping criterion is met.

A recent addition to sequential feature selection is the sequential LASSO cum EBIC in ultra high-dimensional feature space (SLasso) by [1], which sequentially solves partially penalized least squares problems and uses EBIC as the stopping criteria. The EBIC proposed by [6] are suitable for model selection in large model spaces. It has the ordinary BIC as a special case. For large model spaces, the ordinary BIC tends to select a model with many spurious variables. Let *k* and k+1 be the number of predictors in two models, respectively. Using EBIC as the selection or stopping criteria, the model with *k* predictors is selected if the EBIC(k + 1) > EBIC(k).

In SLasso, sequentially solving the partially penalized least squares reduces to selecting the feature(s) which maximize the ordinary correlation coefficient between the features and the response variable at each step. It is well-known that the Pearson correlation coefficient is used for measuring the strength of linear associations. Thus, maximizing the Pearson correlation coefficient might not work well for data structures where the relationship between at least one feature and the response variable is nonlinear.

In this article, we propose the use of the energy distance correlation instead of the ordinary correlation coefficient to identify and maximize both the linear and nonlinear relationships that might exist between each feature and the response. Energy distance is a metric that measures the distance between the distributions of random vectors. The name ‘energy’ is motivated by analogy to the potential energy between objects in a gravitational space. The potential energy is zero if and only if the locations (the gravitational centers) of the two objects coincide, and increases as their distance in space increases [7]. The energy distance correlation has an explicit relationship with the product-moment correlation, but unlike the classical definition of correlation, energy distance correlation is zero only if the random vectors are independent. The empirical energy distance correlation is based on Euclidean distances between sample elements rather than sample moments.

The remainder of the article is arranged as follows: in Section 2, we discuss the derivation of the energy distance correlation, extended Bayesian information criteria and our proposed method (energy distance correlation with EBIC (Edc + EBIC)). In Section 3, we report simulation studies comparing Edc + EBIC with various other methods and provide an analysis of real data. In Section 4, we conclude the article with a discussion of the results.

## 2. EBIC with Energy Distance Correlation

### 2.1. Energy Distance Correlation

The authors in [8] proposed the energy distance correlation between two random variables. Suppose that $W\in R^{p}$ and $Z\in R^{q}$ are two random vectors with E∥W∥<∞, and E∥Z∥<∞, where ∥.∥ is the euclidean norm and E is the expected value. Let *F* and *G* be the cumulative distribution function (CDF) of *W* and *Z*, respectively. Further, let $W^{{}^{\prime }}$ denote an independent and identically distributed (iid) copy of *W*; that is, *W* and $W^{{}^{\prime }}$ are iid. Similarly, *Z* and $Z^{{}^{\prime }}$ are iid.

The squared energy distance can be defined in terms of expected distances between the random vectors
$$D^{2}\left(F,G\right):=2E∥W-Z∥-E∥W-W^{\prime }∥-E∥Z-Z^{\prime }∥\geq 0,$$
and the energy distance between distributions *F* and *G* is defined as the square root of $D^{2}\left(F,G\right)$.

The energy distance correlation between random vectors *W* and *Z* with finite first moments is the nonnegative number R(W,Z) defined by
$$R\left(W,Z\right)=\left\{\begin{array}{cc} \frac{\nu^{2}\left(W,Z\right)}{\sqrt{\nu^{2}\left(W\right)\nu^{2}\left(Z\right)}}, & \nu^{2}\left(W\right)\nu^{2}\left(Z\right)>0\\ 0, & \nu^{2}\left(W\right)\nu^{2}\left(Z\right)=0 \end{array}\right.$$
where $\nu^{2}\left(W,Z\right)$ is the energy distance covariance between *W* and *Z*, $\nu^{2}\left(W\right)$ and $\nu^{2}\left(Z\right)$ are the energy distance variance of *W* and *Z* respectively.

For a statistical sample $\left(w,z\right)=\{\left(w_{k},z_{k}\right),k=1,2,\ldots ,n\}$ from a pair of real-valued or vector-valued random variables (W,Z), the sample energy distance correlation, $R_{n}\left(W,Z\right)$, is calculated by first computing the *n* by *n* distance matrices $\left(a_{j,k}\right)$ and $\left(b_{j,k}\right)$ containing all pairwise distances $\left(a_{j,k}\right)=∥W_{j}-W_{k}∥,$
j,k=1,2,…,n and $\left(b_{j,k}\right)=∥Z_{j}-Z_{k}∥,$
j,k=1,2,…,n where ∥.∥ denotes euclidean norm. Secondly, calculate all doubly centered distances $A_{j,k}=a_{j,k}-{\bar{a}}_{j.}-{\bar{a}}_{.k}+{\bar{a}}_{..},\;B_{j,k}=b_{j,k}-{\bar{b}}_{j.}-{\bar{b}}_{.k}+{\bar{b}}_{..}$ where ${\bar{a}}_{j.}$ is the $j^{th}$ row mean, ${\bar{a}}_{.k}$ is the $k^{th}$ column mean, and ${\bar{a}}_{..}$ is the grand mean of the distance matrix of the *w* sample. The notation is similar for the *b* values.

The squared sample distance covariance (a scalar) is the arithmetic average of the products $A_{j,k}B_{j,k}.$
$$\nu_{n}^{2}\left(w,z\right)=\frac{1}{n^{2}}\sum_{j=1}^{n}\sum_{k=1}^{n}A_{j,k}B_{j,k}.$$

The sample energy distance variance for sample *w*
$$\nu_{n}^{2}\left(w\right)=\frac{1}{n^{2}}\sum_{j,k=1}^{n}A_{j,k}^{2}.$$

The sample energy distance variance for sample *z*
$$\nu_{n}^{2}\left(z\right)=\frac{1}{n^{2}}\sum_{j,k=1}^{n}B_{j,k}^{2}.$$

The sample energy distance correlation is
$$R_{n}\left(W,Z\right)=\left\{\begin{array}{cc} \frac{\nu_{n}^{2}\left(W,Z\right)}{\sqrt{\nu_{n}^{2}\left(W\right)\nu_{n}^{2}\left(Z\right)}}, & \nu_{n}^{2}\left(W\right)\nu_{n}^{2}\left(Z\right)>0\\ 0, & \nu_{n}^{2}\left(W\right)\nu_{n}^{2}\left(Z\right)=0. \end{array}\right.$$

Some basic properties of the distance correlation are as follows:(i)$0\leq R_{n}\left(W,Z\right)\leq 1;$(ii)If $E(|W{|}_{p}+|Z{|}_{q})<\infty ,$ then $R_{n}\left(W,Z\right)=0$ if and only if *W* and *Z* are independent;(iii)Suppose that $R_{n}\left(W,Z\right)=1$. Then, there exist a vector *a*, a nonzero real number *b* and an orthogonal matrix *C* such that Z=a+bWC.

For further details on the energy distance correlation, see [7].

### 2.2. EBIC

Ref. [6] derived the EBIC which have special cases as AIC and BIC. Let $\left\{\left(y_{i},x_{i}\right):i=1,2,\ldots ,n\right\}$ be independent observations. Suppose that the conditional density function of $y_{i}$ given $x_{i}$ is $f(y_{i}|x_{i},\beta ),$ where $\beta \in \Theta \subset R^{p_{n}},p_{n}$ being a positive integer. The likelihood function of β is given by
$$L_{n}\left(\beta \right)=f\left(x;\beta \right)=\prod_{i=1}^{n}f\left(y_{i}|x_{i},\beta \right).$$
Denote $y={\left(y_{1},y_{2},\ldots ,y_{n}\right)}^{\tau }$. Let $s\subset \{1,2,\ldots ,p_{n}\}$ and β(s) be the parameter vector β with those components outside *s* set to 0. Let *S* be the underlying model space, i.e., $S=\{s:s\subseteq \left\{1,2,\ldots ,p_{n}\right\}\},$ let p(s) be a prior for model *s*. Assume that, given *s*, the prior density of β(s) is π(β(s)). The posterior is
$$p\left(s|y\right)=\frac{m(y|s)p(s)}{\sum_{s\in S}m\left(y|s\right)p\left(s\right)},$$
where m(y|s) is the likelihood in model *s*, i.e.,
m(Y|s)=∫f(y;β(s))π(β(s))dβ(s).
Suppose *S* is partitioned into $\cup_{j=1}^{p}S_{j},$ such that models within each $S_{j}$ have an equal dimension. Let $\tau (S_{j})$ be the size of $S_{j}$. Assign the prior distribution $P(S_{j})$ proportional to $\tau^{\eta }\left(S_{j}\right)$ for some η between 0 and 1. For each $s\in S_{j}$, assign equal probability, $p\left(s|S_{j}\right)=1/\tau \left(S_{j}\right)$; this is equivalent to P(s) for $s\in S_{J}$ proportional to $\tau^{-\gamma }\left(S_{j}\right)$, where γ=1−η. Then, the extended BIC family is given by
$$EBIC_{\gamma }\left(s\right)=-2\log L_{n}\left\{\hat{\beta }\left(s\right)\right\}+\left|s\right|\log\left(n\right)+2\gamma \ln\left(\tau \left(S_{\left|s\right|}\right)\right),0\leq \gamma \leq 1,$$
where $\hat{\beta }\left(s\right)$ is the maximum likelihood estimator of β(s) and |s| is the number of components in *s*.

### 2.3. Energy Distance Correlation with EBIC (Edc + EBIC) Algorithm

We propose a sequential model selection method which we call energy distance correlation with EBIC, and for convenience abbreviate it as Edc + EBIC. Let $y_{i},i=1,\ldots ,n$ be a continuous response variable and $x_{j},j=1,\ldots ,p$ be an n×p data matrix. Let *S* be the index set of all predictors. Let $s_{0}=\left\{j:\beta_{j}\neq 0,j=1,\ldots ,p\right\}$. For s⊂S, let $s^{-}=s^{c}\cap s_{0}$. If $s\subset s_{0}$, then $s^{-}$ is the complement of *s* in $s_{0}$. Let $p_{0}=\left|s_{0}\right|$ be the number of elements in the set $s_{0}$.

At the initial stage we standardize all the variables. Next, we find the energy distance correlation between the response variable and each of the predictor variables—$\left\{R\left(x_{j},y\right)\;j=1,\ldots ,p.\right\}$. We then select the predictor (feature) which has the highest distance correlation with the response and store it in the active set $s_{*1}$.

Let L(s) be the linear space spanned by the columns of X(s) and H(s) its corresponding projection matrix, i.e., $H\left(s\right)=X\left(s\right){\left[X^{\tau }\left(s\right)X\left(s\right)\right]}^{-1}X^{\tau }\left(s\right)$. Next, we compute $I-H(s_{*1})$, $\mathrm{EBIC}(s_{*1})$, $\tilde{y}=\left[I-H\left(s_{*k}\right)\right]y\;\mathrm{and}\;{\tilde{x}}_{j}=\left[I-H\left(s_{*k}\right)\right]x_{j}.$ The variable $\tilde{y}$ is the unexplained part of *y* by $X(s_{*1})$. This gives $X(s_{*1})$ close to a zero chance of being selected in the subsequent steps.

For the general step where k>1, we calculate $\left\{R\left({\tilde{x}}_{j},\tilde{y}\right)\;j=1,\ldots ,p.\right\}$ and update the active set to $s_{*k+1}$, which is the union of all the previous selected variables and the current one. We then compute $\mathrm{EBIC}(s_{*k+1})$ and compare it with $\mathrm{EBIC}(s_{*k})$. The procedure stops if $\mathrm{EBIC}\left(s_{*k+1}\right)>\mathrm{EBIC}\left(s_{*k}\right)$. The selected variables which we call the relevant variables will be $X(s_{*k})$. We can then fit a linear regression model between the response *y* and the relevant variables.

We wish to note that care must be taken in fitting this model because some of the predictors might be non-linearly related to *y*, and thus some of the predictors may have to enter into the model in their quadratic or cubic form, etc. Alternatively, a Box–Cox transformation can be performed on the data before fitting the model.

The algorithm details are given in the following.

*Initial Step:* With $y,x_{j},j=1,\ldots ,p$ satisfying $y^{\tau }1=0,x_{j}^{\tau }1=0$ and $y^{\tau }y=n,x_{j}^{\tau }x_{j}=n$, compute $R(x_{j},y)$ for j∈S. Let
$$s_{TEMP}=\left\{j:R\left(x_{j},y\right)=\max_{j^{\prime }\in S}R\left(x_{j^{\prime }},y\right)\right\}.$$Let $s_{*1}=s_{TEMP}$ be the active set. Compute $I-H(s_{*1})$ and $\mathrm{EBIC}(s_{*1}),$ where $H\left(s\right)=X\left(s\right){\left[X^{\tau }\left(s\right)X\left(s\right)\right]}^{-1}X^{\tau }\left(s\right).$*General Step:* In the selection step *k*, compute $R({\tilde{x}}_{j},\tilde{y})$ for $j\in s_{*k}^{c},$ where $\tilde{y}=\left[I-H\left(s_{*k}\right)\right]y,{\tilde{x}}_{j}=\left[I-H\left(s_{*k}\right)\right]x_{j}.$ Let
$$s_{TEMP}=\left\{j:R\left({\tilde{x}}_{j},\tilde{y}\right)=\max_{j^{\prime }\in S_{*k}^{c}}R({\tilde{x}}_{j},\tilde{y})\right\}.$$Let $s_{*k+1}=s_{*k}\cup s_{TEMP}$. Compute $\mathrm{EBIC}(s_{*k+1})$. If $\mathrm{EBIC}\left(s_{*k+1}\right)>\mathrm{EBIC}\left(s_{*k}\right)$, stop; otherwise, continue computing $I-H(s_{*k+1})$.When the process terminates, return the least-squares estimates for parameters in the selected model.

### 2.4. Selection Consistency of Edc + EBIC

We attempt to establish the large sample property for the Edc + EBIC. We will show that under regular conditions, the Edc + EBIC is selection-consistent. The proof essentially follows the approach in [9]. We proceed with the regularity conditions.

**Assumption 1.** 
*Random vectors X and Y possess the subexponential tail probabilities, uniformly in p, specified as follows. There is a constant $a_{0}>0$, such that for any $0<a\leq 2a_{0},$$\sup_{p}\max_{1\leq k\leq p}E\left\{\exp\left(a{∥X_{k}∥}_{1}^{2}\right)\right\}<\infty$ and $E\{\exp\left(a{∥Y∥}_{q}^{2}\right)\}<\infty$.*


**Assumption 2.** 
*The minimum distance correlation of predictors on which y functionally depends satisfies $\min_{j\in s_{0}}R\left({\tilde{X}}_{j},\tilde{Y}\right)\geq 2cn^{-d},$ for some constants 0<c<1 and 0≤d<1/2.*


**Assumption 3.** 
*For the index set S of all predictors, let $s_{0}=\left\{j:\beta_{j}\neq 0,j=1,\ldots ,p\right\}$ and $p_{0}=\left|s_{0}\right|$ ($p_{0}$ is the number of elements in the set $s_{0}$). For s⊂S let $s^{-}=s^{c}\cap s_{0}$. If $s\subset s_{0}$ then $s^{-}$ is the complement of s in $s_{0}$. For $s\subset s_{0}$, $\max_{j\in s_{0}^{c}}R\left({\tilde{X}}_{j},\tilde{Y}\right)<q\max_{j\in s^{-}}R\left({\tilde{X}}_{j},\tilde{Y}\right)$ for some 0<q<1, where $\tilde{Y}=\left[I-H\left(s_{*k}\right)\right]Y,\;{\tilde{X}}_{j}=\left[I-H\left(s_{*k}\right)\right]X_{j}.$ For k=0, $s_{*0}$ is defined as the empty set ∅.*


Details for requiring Assumption 1 and 2 are stated in [9]. Intuitively, Assumption 1 is required to make it easy to establish a relationship between the energy distance correlation and the squared Pearson correlation to aid with the derivations in the proof. Assumption 2 requires that the energy distance correlation for the relevant predictors cannot be too small. Assumption 3 requires that the maximum energy distance correlation between the selected features and the residual response $\tilde{Y}$ is smaller than the maximum energy distance correlation between the remaining features and the residual response in the sequential step of the algorithm.

**Theorem 1.** 
*Suppose that Assumptions 1–3 hold. The proposed Edc + EBIC with the energy distance correlation is consistent, i.e.,*

$$\lim_{n\to \infty }P\left(s_{*k^{*}}=s_{0n}\right)=1,$$

*where $s_{*k^{*}}$ is the set of features selected at the $k^{*th}$ step of Edc + EBIC such that $|s_{*k^{*}}|=p_{0n,}$$s_{0n}$ is the set of relevant features and $p_{0n}=\left|s_{0n}\right|$.*


**Proof.** Suppose that $X\in R^{p}$ and $Y\in R^{q}$ with cumulative distribution function (CDF) *F* and *G*, respectively, where E∥X∥<∞, and E∥Y∥<∞. The energy distance correlation R(X,Y) is the square root of the standardized coefficient:
$$R\left(X,Y\right)=\left\{\begin{array}{cc} \frac{\nu^{2}\left(X,Y\right)}{\sqrt{\nu^{2}\left(X\right)\nu^{2}\left(Y\right)}}, & \nu^{2}\left(X\right)\nu^{2}\left(Y\right)>0\\ 0, & \nu^{2}\left(X\right)\nu^{2}\left(Y\right)=0 \end{array}\right.$$
where 0≤R(X,Y)≤1. In the numerator is the distance covariance defined by [8], as
$$dcov^{2}\left(x,y\right)=S_{1}+S_{2}-2S_{3},$$
where $S_{j},j=1,2,\mathrm{and}\;3$ are defined as:
(2)$$\begin{array}{cc} S_{1}= & E∥X-X^{\prime }∥∥Y-Y^{\prime }∥\\ S_{2}= & E∥X-X^{\prime }∥E∥Y-Y^{\prime }∥\\ S_{3}= & E∥X-X^{\prime }∥∥Y-Y^{\prime \prime }∥ \end{array}$$
where $\left(X,Y\right),\left(X^{\prime },Y^{\prime }\right)$, and $\left(X^{\prime \prime },Y^{\prime \prime }\right)$ are independently and identically distributed.For a random sample $\left\{\left(x_{i},y_{i}\right),i=1,\ldots ,n\right\}$ from (x,y), [8] estimated $S_{1},S_{2},S_{3}$ as:
$$\begin{array}{cc} \hat{S_{1}}= & \frac{1}{n^{2}}\sum_{k,l=1}^{n}|x_{k}-x_{l}{|}_{p}{\left|y_{k}-y_{l}\right|}_{q}\\ \hat{S_{2}}= & \frac{1}{n^{2}}\sum_{k,l=1}^{n}|x_{k}-x_{l}{|}_{p}\frac{1}{n^{2}}\sum_{k,l=1}^{n}{\left|y_{k}-y_{l}\right|}_{q}\\ \hat{S_{3}}= & \frac{1}{n^{3}}\sum_{k=1}^{n}\sum_{l,m=1}^{n}|x_{k}-x_{l}{|}_{p}{\left|y_{k}-y_{l}\right|}_{q} \end{array}$$
so the sample distance covariance is ${\hat{dcov}}^{2}=\hat{S_{1}}+\hat{S_{2}}-2\hat{S_{3}}$.The remaining part of the proof is to show that the energy distance correlation is uniformly consistent and has the sure screening property. The numerator and denominator of the energy distance correlation are similar, so to show the uniform consistency of the energy distance correlation it suffices to show that both the numerator and the denominator are uniformly consistent.The uniform consistency of the numerator, ${\hat{dcov}}^{2}=\hat{S_{1}}+\hat{S_{2}}-2\hat{S_{3}}$, of the energy distance correlation between the random vectors (x,y) is shown by [9]. However, in the general step of the sequential algorithm for Edc + EBIC, the energy distance correlation is calculated between the residuals $\tilde{y}=\left[I-H\left(s_{*k}\right)\right]y,\;\mathrm{and}\;{\tilde{x}}_{j}=\left[I-H\left(s_{*k}\right)\right]x_{j}$ at each step of the algorithm. Thus, to show the uniform consistency of Edc+EBIC it is equivalent to follow the proof by [9].Additionally, in [9] they showed that the energy distance correlation has the sure screening property. They showed that the energy distance is able to select a subset of the features which contains the relevant features. Their argument applies here because we used the energy distance correlation as well, thus the Edc + EBIC has the sure screening property.Therefore, the Edc + EBIC is selection-consistent since it is uniformly consistent and has the sure screening property. The proof is complete. □

## 3. Simulation Studies and Data Analysis

### 3.1. Sure Independence Screening Using Energy Distance Correlation

We establish the need for Ebc + EBIC by firstly examining the performance of a sure independence screening method introduced by [9] called Distance Correlation Sure Independence Screening (DC-SIS). This is similar to the Sure Independence Screening (SIS) introduced by [10].

In SIS, they perform a componentwise regression between each predictor and the response and select the first n−1or[n/log(n)] predictors with the largest estimates. Performing a componentwise regression is equivalent to finding the ordinary correlation between the response and each predictor when the two variables are standardized. Hence, in DC-SIS, they replaced the ordinary correlation with the energy distance correlation.

We examine the performance of DC-SIS through a simulation study. We are interested in observing, on average, the model size selected by SCAD or ALasso if we screened the data first using DC-SIS. We present two simulation set-ups. For each simulation we generated two hundred datasets, and for each dataset we ran SCAD, ALasso, DC-SIS + SCAD, DC-SIS + ALasso and found the average model size and the standard deviation.

In [10], details of two simulation setups we adapted for this subsection are discussed, namely independent features setup and dependent features setup. In Table 1 and Table 2 we present results under the independent features setup and in Table 3, Table 4 and Table 5 we present results under the dependent features setup. In each simulation, *n* is the sample size, *p* is the number of features and *s* is the true model size. For the screening using the energy distance correlation we chose d=[n/logn] features and applied SCAD or ALasso.

In Table 1 and Table 2, we report the average selected model size and their standard deviations. We observe that applying the sure screening by distance correlation before either SCAD or ALasso in all cases did not lead to significant differences in the average model size when SCAD and ALasso were applied directly to the data. This suggests that either applying distance correlation before SCAD or ALasso did not yield the intended result, and thus needs some improvement.

In Table 3, Table 4 and Table 5 we report the selected model size and the standard deviation. We observe that applying DC-SIS followed by either SCAD or ALasso did not yield any significant difference in the average model size, as was also observed in the independent features setup.

### 3.2. Simulation Studies to Compare Edc + EBIC with Other Feature Selection Methods

In this simulation study we adopted two simulation setups from [1], which they call group A and group B, respectively. Under their group A we considered four settings of the covariance structure for the design matrix X, namely GA1, GA2, GA3, and GA5. In their group B setup we considered all three settings of the covariance structure for the design matrix X, namely GB1, GB2, and GB3. We compared the performance of adaptive LASSO (ALasso) [11], SCAD [12], SIS+SCAD [10], SLasso [1], and the energy distance correlation with EBIC (Edc + EBIC) based on the model size (MSize), positive discovery rate (PDR), $\mathrm{PDR}=\frac{|s_{*k^{*}}\cap s_{o}|}{|s_{o}|}$, and false discovery rate, $\mathrm{FDR}=\frac{|s_{*k^{*}}\cap s_{0}^{c}|}{|s_{*k^{*}}|}$ averaged over 200 and 500 simulations, respectively.

We considered the diverging pattern $\left(n,p,p_{o}\right)=\left(n,\left[5e^{n^{0.3}}\right],\left[4n^{0.16}\right]\right)$, meaning that as the sample size increased, the number of predictors increased and the number of relevant predictors also increased. The coefficients were generated as independent random variables distributed as ${\left(-1\right)}^{u}(4n^{-0.15}+|z|),$ where u∼PBernoulli(0.4) and *z* is a normal random variable with mean 0 and satisfies P(|z|≥0.1)=0.25. The variance of the error term in the linear model was determined by
$$h=\frac{\beta^{\tau }\Sigma \beta }{\beta^{\tau }\Sigma \beta +\sigma^{2}}=0.8$$
where Σ is the variance-covariance matrix of relevant features. The response variable is simulated from the sparse high-dimensional regression (SHR) model
$$y_{i}=\beta_{0}+\sum_{j=1}^{p}\beta_{j}x_{ij}+\epsilon_{i},i=1,...,n$$

#### 3.2.1. Group A Simulations and Results

We used two sample sizes, n=100 and n=200. By the diverging pattern considered for the simulation, for a sample size of 100 we have $\left(n,p,p_{0}\right)=\left(100,268,8\right)$ and for a sample size of 200 we have $\left(n,p,p_{0}\right)=\left(200,672,9\right)$. Under the two sample sizes we had eight and nine relevant features, respectively, and we expected a well-performed selection model to select the right number of relevant predictors (model size) on average. The details about the covariance structure for GA1, GA2, GA3, and GA5 are in [1].

In Table 6 and Table 7, we report the simulation results under the conditions for GA1, GA2, GA3, and GA5 using sample sizes 100 and 200, respectively. We observed that under all setups, Edc + EBIC improved in terms of the average model size, PDR, and FDR, as we increased the sample size. This demonstrates the selection consistency of Edc + EBIC.

#### 3.2.2. Group B Simulations and Results

We considered three different covariance structures named GB1, GB2, and GB3 for the features (predictors), as used in [1]. We also increased the signal-to-noise ratio by increasing the value of the expected predictors.

GB1. All the features had constant pairwise correlation $p_{ij}=0.5.\left(n,p,p_{0}\right)=\left(100,200,15\right).$σ=1.5. The coefficients of the relevant features were specified as $|\beta_{j}|=2.5$ for 1≤j≤5,1.5 for 6≤j≤10,0.5 for 11≤j≤15. The signs of the coefficients were determined as ${\left(-1\right)}^{u_{i}}$, where the $u_{i}$s were iid Bernoulli random variables with probability of success p=0.5.

GB2. This structure was the same as in GB1, that is, $\left(n,p,p_{0}\right)=\left(100,200,15\right)$ and σ=1.5. The covariance structure of the features was specified such that the partially orthogonality condition [11] was satisfied. Specifically, while $s_{0}$ was taken as {1,…5,11,…,15,21,…,25}, the correlations were specified as $\rho_{ij}=0.5^{\left|i-j\right|}$ for 1≤i≤215 and 1≤j≤215. The coefficients were specified as |β|=2.5 for 1≤j≤5,1.5 for 10≤j≤15,0.5 for 21≤j≤25. The signs of the coefficients were determined in the same way as in GB1.

GB3. $\left(n,p,p_{0}\right)=\left(100,1000,10\right)$ and σ=1. The relevant features were generated as iid standard normal variables with coefficients (3, 3.75, 4.5, 5.25, 6, 6.75, 7.5, 8.25, 9, 9.75). The irrelevant features were generated as
$$x_{j}=0.25Z_{j}+\sqrt{0.75}\sum_{k\in s_{0}}X_{k},j\notin s_{0},$$
where $Z_{j}s$ are iid standard normal and independent from the relevant features.

In Table 8, we report the simulation results under the conditions for GB1, GB2, and GB3. We observed that SLasso and Edc + EBIC performed better. SLasso had the highest PDR while Edc + EBIC had the lowest FDR. Thus, when there was some correlation among the features, Edc + EBIC still performed well.

#### 3.2.3. Real Data Example

The new method was applied to the gene expression data used in [1]. For the data and details of data collection and variable definitions, see [1].

This study aimed to find the probes among the remaining 18,975 probes most closely related to TRIM32. The response variable was the expression level of probe 1389163_at. The features were the expression levels of the remaining 18,975 probes. Of the 18,975 probes, the top 3000 probes with the largest variances were considered. The expression levels were standardized to have mean 0 and standard deviation 1.

In our analysis of the data, for each of the 100 replications we selected a random sample of size 100 from 120 rats and applied Edc + EBIC to the sample. From these 100 replications, Edc + EBIC selected the distinct probes 1367705_at and 1367728_at. In [1], the probes selected by five (5) variable selection methods are reported. The probes selected by Edc + EBIC did not intersect with any of the probes these methods selected. Among the probes selected by these five (5) methods, some intersected, but this is not a surprise because these methods essentially maximized only the linear relationship between TRIM32 and each of the probes. Since Edc + EBIC is capable of detecting and maximizing both the linear and nonlinear relationships that might exist between TRIM32 and each of the probes, as evidenced in the simulation studies by the high PDR and low FDR, we are convinced that the two probes selected by our method are the most associated with TRIM32.

## 4. Conclusions and Discussion

From the simulation results in Table 6 and Table 7 we observed that, as the sample size (n) increased, Edc + EBIC selected on average the expected number of predictors and did so with decreasing standard deviations, meaning that through the simulation runs more and more of the selected predictors were close to the expected number of relevant predictors. We also observed the positive discovery rate of 100%, indicating that on average for each simulation run, out of the selected features all of the relevant features were selected. Of greater importance was the small false discovery rates recorded as the sample size increased.

We found through simulation studies that when we applied the Energy Distance Correlation Sure Independence Screening proposed by [9] for variable screening followed by a regularization method such as SCAD and ALasso, the average model size selected was higher than expected and with high standard deviations.

## Figures and Tables

**Table 1 entropy-25-00014-t001:** Comparing model size selected with or without screening for n=200,s=8,p=1000.

Methods	MSize (SD)
SCAD	12.87 (7.292)
DC-SIS + SCAD	10.7 (3.1575)
ALasso	25.24 (9.0365)
DC-SIS + ALasso	11.74 (4.419)

**Table 2 entropy-25-00014-t002:** Comparing model size selected with or without screening for n=800,s=14,p=3000.

Methods	MSize (SD)
SCAD	16.62 (2.78807)
DC-SIS + SCAD	16.69 (3.5525)
ALasso	14.78 (0.7860)
DC-SIS + ALasso	14.78 (3.8522)

**Table 3 entropy-25-00014-t003:** Comparing model size selected with or without screening for n=200,p=1000,s=5.

Methods	MSize (SD)
SCAD	12.215 (12.2560)
DC-SIS + SCAD	7.335 (2.6109)
ALasso	44.485 (13.8041)
DC-SIS + ALasso	8.21 (2.5844)

**Table 4 entropy-25-00014-t004:** Comparing model size selected with or without screening for n=200,p=1000,s=8.

Methods	MSize (SD)
SCAD	14.625 (10.3324)
DC-SIS + SCAD	10.905 (2.3158)
ALasso	19.95 (3.2789)
DC-SIS + ALasso	12.36 (3.9875)

**Table 5 entropy-25-00014-t005:** Comparing model size selected with or without screening for n=800,p=3000,s=14.

Methods	MSize (SD)
SCAD	19.185 (5.0146)
DC-SIS + SCAD	17.675 (4.6038)
ALasso	38.125 (5.6372)
DC-SIS + ALasso	31.845 (13.1011)

**Table 6 entropy-25-00014-t006:** We compared the methods using PDR, FDR, and model size (MSize) averaged over 200 simulation replications. The relevant predictors were 8 and the sample size was 100. The standard deviations are in parentheses.

Setting	Methods	MSize	PDR	FDR
GA1	ALasso	34.105 (13.95)	1.000 (0.000)	0.721 (0.120)
	SCAD	25.735 (5.020)	1.000 (0.000)	0.676 (0.065)
	SIS + SCAD	8.100 (1.790)	0.866 (0.239)	0.157 (0.167)
	SLasso	8.565 (0.848)	1.000 (0.000)	0.058 (0.081)
	Edc + EBIC	8.365 (1.375)	0.978 (0.125)	0.056 (0.085)
GA2	ALasso	34.455 (11.095)	1.000 (0.000)	0.754 (0.108)
	SCAD	25.650 (6.720)	0.876 (0.141)	0.709 (0.075)
	SIS + SCAD	7.335 (1.740)	0.813 (0.182)	0.103 (0.107)
	SLasso	6.055 (1.725)	0.688 (0.185)	0.080 (0.104)
	Edc + EBIC	6.075 (1.713)	0.717 (0.195)	0.050 (0.082)
GA3	ALasso	14.710 (3.847)	1.000 (0.000)	0.423 (0.131)
	SCAD	26.27 (5.244)	1.000 (0.000)	0.680 (0.070)
	SIS + SCAD	8.165 (1.160)	0.951 (0.113)	0.062 (0.078)
	SLasso	8.625 (1.005)	1.000 (0.000)	0.062 (0.089)
	Edc + EBIC	8.265 (1.373)	0.976 (0.132)	0.048 (0.074)
GA5	ALasso	23.845 (7.005)	0.964 (0.057)	0.652 (0.092)
	SCAD	24.070 (6.147)	0.997 (0.020)	0.642 (0.102)
	SIS + SCAD	7.605 (2.020)	0.832 (0.245)	0.127 (0.141)
	SLasso	7.650 (2.182)	0.856 (0.217)	0.089 (0.113)
	Edc + EBIC	7.180 (2.453)	0.842 (0.270)	0.050 (0.081)

**Table 7 entropy-25-00014-t007:** We compared the methods using PDR, FDR, and model size (MSize) averaged over 200 simulation replications. The relevant predictors were 9 and the sample size was 200. The standard deviations are in parentheses.

Setting	Methods	MSize	PDR	FDR
GA1	ALasso	27.670 (12.996)	1.000 (0.000)	0.638 (0.180)
	SCAD	17.035 (7.746)	1.000 (0.000)	0.454 (0.168)
	SIS + SCAD	9.215 (1.507)	1.000 (0.000)	0.112 (0.123)
	SLasso	8.710 (0.944)	1.000 (0.000)	0.072 (0.088)
	Edc + EBIC	8.42 (0.712)	1.000 (0.000)	0.045 (0.071)
GA2	ALasso	27.92 (9.686)	1.000 (0.000)	0.675 (0.124)
	SCAD	15.11 (6.241)	1.000 (0.000)	0.397 (0.171)
	SIS + SCAD	9.16 (1.509)	1.000 (0.000)	0.108 (0.171)
	SLasso	8.72 (0.947)	1.000 (0.000)	0.073 (0.089)
	Edc + EBIC	8.49 (0.763)	1.000 (0.000)	0.051 (0.076)
GA3	ALasso	27.115 (12.867)	1.000 (0.000)	0.632 (0.177)
	SCAD	16.245 (7.770)	1.000 (0.000)	0.434 (0.162)
	SIS + SCAD	9.22 (1.617)	1.000 (0.000)	0.110 (0.128)
	SLasso	8.70 (0.857)	1.000 (0.000)	0.072 (0.084)
	Edc + EBIC	8.47 (0.694)	1.000 (0.000)	0.050 (0.071)
GA5	ALasso	38.95 (8.308)	0.939 (0.075)	0.797 (0.054)
	SCAD	19.075 (7.427)	1.000 (0.000)	0.520 (0.159)
	SIS + SCAD	9.975 (1.858)	1.000 (0.000)	0.174 (0.132)
	SLasso	8.765 (1.125)	0.989 (0.061)	0.087 (0.094)
	Edc + EBIC	8.44 (0.768)	0.998 (0.025)	0.048 (0.073)

**Table 8 entropy-25-00014-t008:** We compared the methods using PDR, FDR, and model size (MSize) averaged over 500 simulation replications. The standard deviations are in parentheses.

Setting	Methods	MSize	PDR	FDR
GB1	ALasso	23.32 (3.018)	0.766 (0.062)	0.501 (0.066)
	SCAD	14.08 (1.644)	0.853 (0.065)	0.085 (0.068)
	SIS + SCAD	10.656 (1.688)	0.694 (0.112)	0.025 (0.067)
	SLasso	14.916 (2.194)	0.893 (0.081)	0.092 (0.089)
	Edc + EBIC	14.094 (2.035)	0.869 (0.088)	0.067 (0.076)
GB2	ALasso	40.474 (11.7331)	0.447 (0.0858)	0.710 (0.0605)
	SCAD	20.966 (7.6121)	0.517 (0.0614)	0.315 (0.1896)
	SIS + SCAD	10.314 (1.0797)	0.403 (0.0427)	0.042 (0.0712)
	SLasso	13.65 (2.038)	0.499 (0.052)	0.077 (0.081)
	Edc + EBIC	14.006 (1.657)	0.67 (0.014)	0.0273 (0.0785)
GB3	ALasso	22.464 (2.4414)	1.000 (0.000)	0.5495 (0.0498)
	SCAD	11.000 (0.000)	1.000 (0.000)	0.091 (0.000)
	SIS + SCAD	9.964 (0.6897)	0.992 (0.0764)	0.107 (0.050)
	SLasso	10.182 (0.475)	0.667 (0.006)	0.015 (0.039)
	Edc + EBIC	10.158 (0.440)	1.000 (0.000)	0.0139 (0.038)

## Data Availability

The data presented in this study is openly available and reported in Luo and Chen (2014) [1] at https://doi.org/10.1080/01621459.2013.877275, accessed on 19 November 2022.

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
