# Peer review of "Feature Selection in High-Dimensional Models via EBIC with Energy Distance Correlation"

_entropy, 2022, doi:10.3390/e25010014_

Round 1
Reviewer 2 Report
The paper deals with the feature selection in high-D linear regression. The new contribu
tion is to use the so-called energy distance correlation in the place of the ordinay cor
relation coefficient to measure the dependence.
How some key information is missing. For example, f_w etc are undefined in v^2(W,Z). Fur
thermore, the norm used in the same formula is undefined either. Though we can guess wha
t they are, how to calculate those measures from data is not clear at all, which could b
e particularly challenging, if ever possible, in the setting of p>n.
The presentation of the paper can benefit tremendously from a careful reading. I list a
few (there are more!) obvious inadequacies below.
In the abstract, the sentence starting with "we propose to use the energy distance corre
lation ..." almost repeats its twice --- delete one.
p.2 l.36: delete ":"
p.2. l.42: results in
p.3 l.53: delete "we are sure"
Round 2
Reviewer 2 Report
The revision has addressed my concerns in the first round. I am happy to the new version published as it is.